# GravIS: mass anomaly products from satellite gravimetry

Christoph Dahle[1], Eva Boergens[1], Ingo Sasgen[2,4], Thorben Döhne[3], Sven Reißland[1], Henryk Dobslaw[1], Volker Klemann[1], Michael Murböck[1,5], Rolf König[1,5], Robert Dill[1], Mike Sips[1], Ulrike Sylla[1], Andreas Groh[3], Martin Horwath[3], and Frank Flechtner[1,5]

[1]Department 1: Geodesy, GFZ German Research Centre for Geosciences, 14473 Potsdam, Germany
[2]Division: Geosciences, Alfred Wegener Institute, 27568 Bremerhaven, Germany
[3]Institut für Planetare Geodäsie, Technische Universität Dresden, 01069 Dresden, Germany
[4]Institute of Geography, University of Augsburg, 86159 Augsburg, Germany
[5]Institute of Geodesy, Technische Universität Berlin, 10623 Berlin, Germany

*Correspondence to*: Christoph Dahle (dahle@gfz-potsdam.de)

**Abstract.** Accurately quantifying global mass changes at the Earth's surface is essential for understanding climate system dynamics and their evolution. Satellite gravimetry, as realized with the Gravity Recovery and Climate Experiment (GRACE) and GRACE Follow-On (GRACE-FO) missions, is the only currently operative remote sensing technique that can track large-scale mass variations, making it a unique monitoring opportunity for various geoscientific disciplines. To facilitate easy

accessibility of GRACE and GRACE-FO (GRACE/-FO in the following) results also beyond the geodetic community, the German Research Centre for Geosciences (GFZ) developed the *Gravity Information Service* (GravIS) portal (https://gravis.gfz-potsdam.de). This work aims to introduce the user-friendly mass anomaly products provided at GravIS that are specifically processed for hydrology, glaciology, and oceanography applications. These mass change data, available in both a gridded representation and as time series for predefined regions, are routinely updated when new monthly GRACE/-FO gravity field

models become available. The associated GravIS web portal visualizes and describes the products, demonstrating their usefulness for various studies and applications in geosciences. Together with GFZ's complementary information portal *globalwaterstorage.info*, GravIS supports widening the dissemination of knowledge about satellite gravimetry in science and society and highlights the significance and contributions of the GRACE/-FO missions for understanding changes in the climate system.

The GravIS products, divided into several data sets corresponding to their specific application, are available at https://doi.org/10.5880/GFZ.GRAVIS_06_L2B (Dahle & Murböck, 2019), https://doi.org/10.5880/COST-G.GRAVIS_01_L2B (Dahle & Murböck, 2020), https://doi.org/10.5880/GFZ.GRAVIS_06_L3_ICE (Sasgen et al., 2019), https://doi.org/10.5880/COST-G.GRAVIS_01_L3_ICE (Sasgen et al., 2020), https://doi.org/10.5880/GFZ.GRAVIS_06_L3_TWS (Boergens et al., 2019), https://doi.org/10.5880/COST-

G.GRAVIS_01_L3_TWS (Boergens et al., 2020a), https://doi.org/10.5880/GFZ.GRAVIS_06_L3_OBP (Dobslaw et al., 2019), and https://doi.org/10.5880/COST-G.GRAVIS_01_L3_OBP (Dobslaw et al., 2020a).

## 1 Introduction

Quantifying the redistribution of mass across the globe is crucial for understanding geophysical processes related to the climate system and its changes. Such mass changes cause spatial and temporal variations of the Earth's gravity field that can be observed by satellite gravimetry. From 2002 to 2017, the Gravity Recovery and Climate Experiment (GRACE; Tapley et al., 2004) mission enabled the generation of high temporal resolution global gravity field models with unprecedented quality. Together with its successor GRACE Follow-On (GRACE-FO; Landerer et al., 2020) launched in 2018, these twin-satellite missions have been providing monthly estimates of the Earth's gravity field with an effective spatial resolution of approx. 300 km on a regular basis for more than 20 years now. From these monthly gravity field solutions, mass changes at the Earth's surface, i.e., changes in the atmosphere, terrestrial hydrosphere, cryosphere, and oceans can be inferred (Wahr et al., 1998). Moreover, GRACE and GRACE-FO (denoted by GRACE/-FO in the following) are also capable of detecting signals of the solid Earth such as megathrust earthquakes and their related co- and post-seismic deformations (e.g., Han et al., 2013) or long-term mass redistributions in the upper mantle related to glacial isostatic adjustment (GIA; e.g., van der Wal et al., 2011). Consequently, both missions contribute substantially to a better understanding of geophysical processes within the Earth's system and its long-term changes due to climate change (Tapley et al., 2019). This fact is also reflected by a continuously increasing number of publications related to GRACE/-FO during the more than two decades of mission lifetime (Fig. 1a). Looking at the different geoscientific disciplines, cryospheric, hydrological, and oceanographic studies and applications contribute most to the overall number of publications, with an increasing ratio for hydrology during the last few years (Fig. 1b).

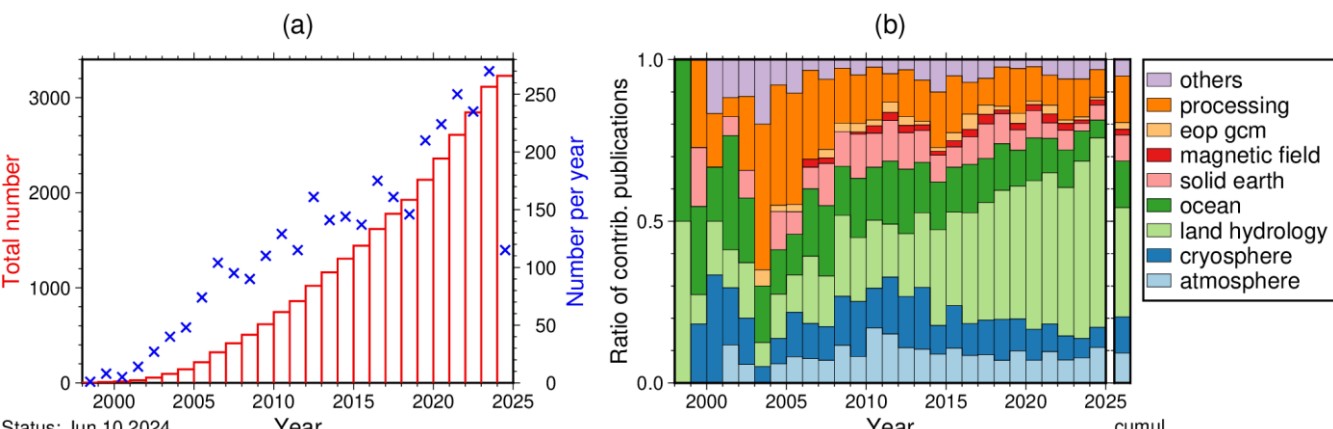

**Figure 1: (a) Number of GRACE/-FO related publications per year and their cumulative distribution. (b) Ratio of these publications contributing to predefined categories; in addition to the different Earth system compartments atmosphere, cryosphere, land hydrology, ocean, and solid earth, applications to the magnetic field, the Earth orientation parameters and geocentre motion, as well as data processing are distinguished. The considered database of GRACE/-FO-related publications is maintained at GFZ and contains currently 3228 peer-reviewed publications (https://www.gfz.de/grace, access date: 10 June 2024).**

To mention just a few of the many significant science results, GRACE/-FO have enabled the observation of seasonal variations of terrestrial water storage (TWS) over the continents, comprising particularly surface water, snow, and soil moisture (e.g., Girotto and Rodell, 2019). The long data record has facilitated the detection and quantification of groundwater depletion, e.g.,

in North India (Chen et al., 2016), droughts (Zhao et al., 2017), or hydroclimatic extreme events in general (Rodell and Li, 2023), but also the assessment of climate change signatures in TWS (Jensen et al., 2019). Furthermore, it has provided essential input to the tracking of large-scale ice-mass loss of the polar ice sheets in Greenland and Antarctica (Shepherd et al., 2012; Groh et al., 2019; Velicogna et al., 2020; Hanna et al., 2024), and of mountain glaciers (Wouters et al., 2019; Sasgen et al., 2022). In the ocean domain, GRACE/-FO has contributed to the estimation of barystatic sea-level rise (Chen et al., 2019), the

global mean sea-level budget (Horwath et al., 2022; Cáceres et al., 2020), and the investigation of general ocean circulation (Johnson and Chambers, 2013; Peralta-Ferriz et al., 2014).

However, the inference of mass changes from gravity field changes requires solving an inverse problem, a common challenge in various geoscientific fields. The problem is non-unique in the sense that for any given gravity field observation, an infinite number of mass distributions exist that satisfy the observations. Moreover, the solution is unstable, meaning that increasing

the spatial resolution is always limited by the uncertainties in the observational data leading to neighbouring solutions that differ vastly. Those challenges are solved (i) by adopting the "thin layer assumption" (Wahr et al., 1998) that presumes that mass variations only take place in an infinitesimal thin layer on the surface of a best-fitting Earth ellipsoid, and (ii) by post-processing the data with spatial smoothing operators that remove small-scale variations (regarded as noise) at the expense of reduced spatial detail. Alternatively, the instability can be overcome with regularization methods during the inversion

procedure, for example by introducing a priori constraints on the spatial or temporal resolution of the target signal, which is the applied strategy when processing so-called mascon solutions.

Analyses typically start from monthly gravity field models represented in terms of spherical harmonic (SH) coefficients (i.e., the GRACE/-FO Level-2 products; spectral domain), commonly provided by GRACE/-FO processing centres. They require knowledge about mitigating errors in the GRACE/-FO data as well as expertise in geophysical signal separation to isolate the

80 mass transport process of interest. To ease access to GRACE/-FO data for scientists without profound geodetic background, efforts have been made to post-process Level-2 products and provide mass anomalies as gridded fields (i.e., GRACE/-FO Level-3 products; spatial domain), e.g., by the GRACE Tellus portal hosted by NASA's Jet Propulsion Laboratory (JPL; Landerer and Swenson, 2012; Chambers and Bonin, 2012). At the same time, directly derived Level-3 products in terms of mascon products have been developed, provided by, e.g., Center for Space Research at the University of Texas (CSR; Save et

al., 2016), Goddard Space Flight Center (GSFC; Loomis et al., 2019a), and JPL (Watkins et al., 2015; Wiese et al., 2016). Advantages and disadvantages of these two Level-3 approaches are discussed in Sect. 5 of this article.

The German Research Centre for Geosciences (GFZ) is coordinating the German contributions to the GRACE-FO satellite mission and is forming the U.S.-German Science Data System (SDS) together with JPL and CSR. As part of these contributions, GFZ developed and permanently maintains the web portal *Gravity Information Service* (GravIS;

https://gravis.gfz-potsdam.de). GravIS aims first to disseminate high-quality and user-friendly Level-3 mass anomaly products

and provide free and easy access to them. Second, by visualizing and describing them, it aims to demonstrate the usefulness of these Level-3 products for studies and applications in hydrology, glaciology, and oceanography. The corresponding data streams are routinely processed and updated in a collaborative effort with partners from the Alfred Wegener Institute, Helmholtz Centre for Polar and Marine Research (AWI) and TUD Dresden University of Technology.

The article is structured as follows: Section 2 gives an overview of the available GravIS products and describes the applied processing steps. In Sect. 3, the GravIS web portal and another portal providing complementary information about GRACE/-FO are introduced and a comparison of GravIS with other similar platforms is provided. Applications of the GravIS products demonstrating their impact on studies of mass change are shown in Sect. 4, followed by a discussion of the limitations of mass anomaly products based on satellite gravimetry in Sect. 5. Data availability including update policies are outlined in Sect.6,

followed by a summary and an outlook on the future evolution of GravIS in Sect. 7.

## 2 Available GravIS mass anomaly products and their processing methods ensuring high data quality

GravIS offers a variety of products related to the GRACE/-FO missions. It distinguishes between data sets in the spectral domain (Level-2B; see Sect. 2.1) and in the spatial domain (Level-3; see Sect. 2.2). One noteworthy feature of GravIS is the generation of dedicated Level-3 products for three different geographical domains: the continental ice sheets, non-glaciated

land surfaces, and the oceans. All Level-3 products are provided as gridded data sets as well as in terms of averages over predefined regions. The following subchapters of this section provide a detailed description of the processing steps of all these different products.

### 2.1 Spectral representation: post-processed gravity field models in terms of spherical harmonic coefficients (Level-2B products)

Level-2B processing starts with monthly GRACE/-FO Level-2 products, i.e., models representing the Earth's gravity field in the spectral domain in terms of SH coefficients. There are two processing chains which are operated in parallel to generate two independent versions of GravIS products:

1) Based on the most recent release of GRACE-/FO Level-2 products processed at GFZ; at the time of writing this article, the most recent release is GFZ RL06 (Dahle et al., 2019b) with the corresponding Level-2 data sets for

GRACE (Dahle et al., 2018) and GRACE-FO (Dahle et al., 2019a).

2) Based on the most recent release of GRACE-FO Level-2 products provided by the Combination Service for Time-variable Gravity fields (COST-G; Jäggi et al., 2020); at the time of writing this article, the most recently released data sets are COST-G RL01 for GRACE (Meyer et al., 2020a) and COST-G RL02 for GRACE-FO, which is a data set similar to Meyer et al. (2020b) with adaptions outlined in Meyer et al. (2024).

The advantage of the GravIS products based on GFZ Level-2 products is given by a shorter latency of the latter compared to COST-G Level-2 products. The rationale behind additionally providing GravIS products based on COST-G is the better

accuracy, which stems from the approach of combining multiple Level-2 products from different analysis centres. These combined gravity field models benefit from the strengths of various processing strategies and show a reduced noise level.

To arrive at SH coefficients that only contain signals caused by mass redistribution, several corrections and reductions are
125 applied to the Level-2 solutions. These post-processed Level-2 coefficients are denoted as Level-2B products and are provided as additional data sets (Dahle and Murböck, 2019; Dahle and Murböck, 2020) for users who wish to undertake mass anomaly inversion by themselves. The following processing sequence from Level-2 to Level-2B is applied in this order:

1) Subtraction of a long-term mean gravity field from the monthly gravity field models to derive anomalies relative to this mean. Currently, i.e., for version 0003 of Dahle and Murböck (2019) and Dahle and Murböck (2020),
respectively, a long-term mean based on the 183 monthly solutions available in the period April 2002 through March 2020 is subtracted.

2) Filtering of the SH coefficients using an anisotropic filter (see Sect. 2.1.1). Several variants of filtered solutions with different filter strengths are generated. This processing step is not mandatory, i.e., there is also one variant with unfiltered solutions.

3) Replacement of particular low-degree SH coefficients (see Sect. 2.1.2).

4) Subtraction of secular trend caused by GIA as provided by a numerical model (see Sect. 2.1.5).

5) Approximation and insertion of SH coefficients of degree 1 to account for geocentre variations (see Sect. 2.1.3).

6) Subtraction of deterministic 161-day periodic signal to account for tidal aliasing errors related to the S2 tide (see Sect. 2.1.4).

**2.1.1 Anisotropic filtering**

Due to the observation geometry which mainly relies on inter-satellite ranging in along-track direction at near-polar orbits, GRACE/-FO gravity field solutions reveal highly anisotropic error characteristics. Thus, filtering is necessary to decorrelate these systematic errors and optimally separate signal and noise in GRACE/-FO Level-2 products. An adequate and widely used filter technique to account for this is the decorrelation method by Kusche et al. (2009), named DDK, which is deduced
from a regularization approach using signal and error information in terms of variance-covariance matrices. The filtering is applied in the spectral domain by multiplying the unfiltered SH coefficients (after subtracting a long-term mean) with a filter matrix. Horvath et al. (2018) adapted this method considering the temporal variations of the error (co)variances. At GravIS, this adapted method, called VDK filter, is used to decorrelate and smooth the monthly Level-2B products. Hence, a tailored filter that explicitly considers the full formal error covariance information of each individual month is applied. Along with
filtering the SH coefficients, the corresponding formal error covariance matrix is also computed, and the SH coefficients' formal uncertainties are modified accordingly.

Figure 2 illustrates the advantages of the VDK method compared to DDK which are most evident for months with insufficient ground track coverage due to short-period repeat orbits (September 2004, May 2012, January and February 2015), but are also clearly visible for other periods such as end of 2016 until mid of 2017 (when one of the two GRACE accelerometers had to be

switched off) or the entire GRACE-FO period (with one accelerometer showing an anomalous behaviour, plus increasing solar activity since 2022). In terms of open ocean root-mean-square (RMS) values, a common metric to assess the error level of monthly GRACE/-FO solutions, a reduction of up to 75 % (January 2015) or even 90 % (February 2015) can be achieved when using the VDK instead of the DDK filter. As already mentioned above, several variants of Level-2B products with different VDK filter strengths, i.e., regularization factors, are generated and made available: VDK1, VDK2, VDK3, VDK4,

VDK5, VDK6, VDK7, and VDK8, respectively, where a smaller number means stronger smoothing and vice versa. In addition to these eight filtered versions of Level-2B, a ninth variant with unfiltered Level-2B products is also provided.

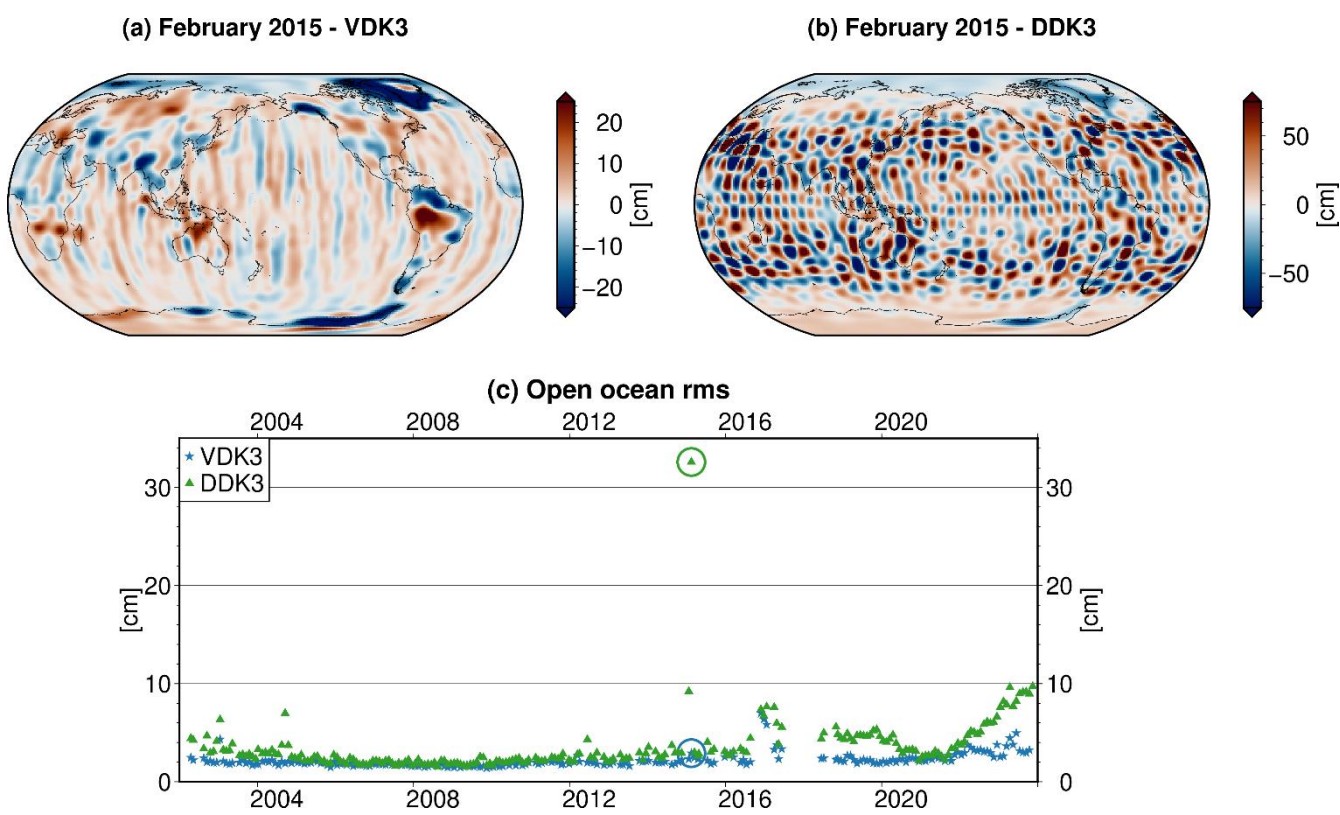

**Figure 2: (a) GFZ RL06 Level-2 solution (max. SH degree 96) for February 2015 in terms of surface mass densities [cm], filtered with VDK3. (b) Same as (a), but filtered with DDK3. (c) Open ocean RMS (distance to coast > 1000 km) for the GFZ GRACE/-FO RL06/RL06.1 time series when filtered with VDK3 (blue asterisks) and DDK3 (green triangles); values for February 2015 are circled.**

### 2.1.2 Replacement of particular low-degree SH coefficients

The SH coefficient of degree 2 and order 0 ($C_{20}$) is related to the flattening of the Earth. Monthly GRACE estimates of C20 are known to be affected by spurious systematic effects (e.g., Cheng and Ries, 2017); for GRACE-FO, the situation is similar. It is thus recommended to replace the native $C_{20}$ coefficients from GRACE/-FO with alternative more reliable estimates. In this context, $C_{20}$ estimates based on satellite laser ranging (SLR) observations to dedicated SLR satellites are commonly used (e.g., Loomis et al., 2019b).

In addition, it was shown by Loomis et al. (2020) that also the $C_{30}$ coefficient, having a rather large impact in particular on Antarctic ice-mass change recovery, is poorly determined from GRACE/-FO whenever accelerometer observations for one of the satellites need to be transplanted from the other one. This is the case for the last months of GRACE (November 2016 through April 2017, June 2017; Bandikova et al., 2019) as well as for the whole GRACE-FO mission (Harvey et al., 2022). For GRACE-FO, however, the availability of improved accelerometer transplant products has significantly improved the

estimation of $C_{30}$ (Behzadpour et al., 2021; Harvey et al., 2024). Consequently, there is no absolute need to replace $C_{30}$ in the most recent GRACE-FO gravity field solutions incorporating one of these improved transplant data sets.

Furthermore, an anomalous behaviour of the $C_{21}$ and $S_{21}$ coefficients is observed for the GFZ GRACE RL06 solutions, particularly during the last seven months of the mission (Dahle et al., 2019b), which can also be mitigated by replacing these two SH coefficients.

In general, the replacement of low-degree SH coefficients during GravIS Level-2B processing is carefully revisited with every new version or release. Currently, the following coefficients including their formal uncertainties are replaced during these periods: $C_{20}$ for the entire GRACE/-FO time series, $C_{30}$ only for the period from November 2016 through June 2017, and $C_{21}/S_{21}$ for the entire GRACE mission but only for the GFZ-based products and not for the products based on COST-G. For replacement, estimates from a combination of GRACE/-FO and SLR processed at GFZ are currently used. The SLR part is

the same as used for GFZ's SLR-only $C_{20}$ time series (König et al., 2019), i.e., it includes observations to the six satellites LAGEOS-1 and -2, AJISAI, Stella, Starlette, and LARES (starting from March 2012) which are processed using the same background models and standards as applied during GFZ's GRACE/-FO Level-2 processing. SLR and GRACE/-FO are then combined on the level of normal equations using relative weights between the individual SLR satellites derived from variance component estimation whereas the relative weighting between SLR and GRACE/-FO is derived empirically. Finally, gravity

field solutions up to degree and order 6 are estimated independently for each month, from which the SH coefficients to be replaced are taken. It is worth mentioning that other replacement time series are also available and that the particular choice of such a time series can significantly impact mass change results (Dobslaw et al., 2020b).

### 2.1.3 Approximation of geocentre variations

The SH coefficients of degree 1 ($C_{10}$, $C_{11}$, $S_{11}$) are related to the distance between the Earth's centre of mass (CM) and centre

of figure (CF), which is commonly denoted as geocentre motion. As satellite missions like GRACE are insensitive to the CF

and thus these coefficients are not estimated but set to zero by definition, no information about geocentre variations is contained in GRACE/-FO Level-2 products. However, such information is essential to correctly quantify oceanic and terrestrial mass distributions. To add this information to the Level-2B products, degree-1 coefficients from an external source are needed. Currently, the approximation method by Swenson et al. (2008) is used at GravIS. As this method is based on monthly GRACE/-FO SH coefficients, it is applied separately for the products based on GFZ and COST-G, respectively, so that dedicated and consistently processed sets of $C_{10}$, $C_{11}$, and $S_{11}$ coefficients can be inserted into each of the two Level-2B product series. Corresponding uncertainties of the degree-1 coefficients are calculated according to Sun et al. (2016) and are used for both the GFZ and the COST-G products.

### 2.1.4 Empirical correction of S2 tidal aliasing errors

The removal of ocean tidal signals from satellite observations is a crucial task during GRACE/-FO Level-2 processing (Sulzbach et al., 2021). However, errors are present in the available global ocean tide models (Stammer et al., 2014), and these errors are expected to be amongst the largest error sources in GRACE-like gravity field recovery (Flechtner et al., 2016). Apart from model errors, additional gravity field errors are caused by temporal aliasing of ocean tide signals related to the space-time sampling of a satellite gravimetry mission (e.g., Murböck et al., 2014).

A prominent alias frequency in the GRACE/-FO gravity fields has a period of 161 days, which is possibly related to model errors of the semi-diurnal solar tide S2 present in both ocean and atmosphere. To mitigate such S2 tidal aliasing errors, an empirical correction is applied during GravIS Level-2B processing: bias, linear trend, annual, semi-annual, and 161-day periodic signals are simultaneously fitted to the time series of monthly Level-2B products and the 161-day periodic signal evaluated at the mid epoch of each month is then subtracted. The estimation of the mentioned deterministic signal components is done individually for the GFZ and COST-G products only once (based on the same period used to calculate the subtracted long-term mean), i.e., the 161-day periodic signal is extrapolated for all subsequent months. A phase offset of 100 degrees between GRACE and GRACE-FO is applied when fitting the S2 tidal aliasing frequency as the nodal planes of both orbits are not specifically aligned to each other (Landerer et al., 2020). Note that the formal uncertainties of the Level-2B coefficients are not considered to change. It also has to be mentioned that the individual Level-2B products are no longer statistically independent from each other after this empirical correction.

Future GRACE/-FO gravity field time series may benefit from reduced ocean tide errors due to advancements in ocean tide models or processing techniques (Abrykosov et al., 2022; Hauk et al., 2023). Consequently, the requirement of correcting S2 tidal aliasing errors for upcoming GravIS product releases will be reassessed.

### 2.1.5 Geophysical correction of signals induced by glacial isostatic adjustment

GIA denotes the surface deformation of both lithosphere and mantle caused by ice-mass changes over the last 100,000 years, which were dominated by the termination of the most recent glacial cycle. Due to the Earth's viscoelastic response to the redistribution of mass between the grounded ice sheets and the fluid ocean, the Earth's gravity field is affected by long-term

secular trends mainly in previously glaciated regions in North America, Fennoscandia, and Antarctica. Moreover, also sea-level is changing both in response to the increase of water stored in the oceans and an adjustment of the local height of the equipotential surfaces of the Earth's gravity field due to attraction effects of the time-variable mass distribution on the continents. Thus, GIA significantly impacts the rate-of-change in SH coefficients of all low degrees and orders.

Currently, the ICE-6G_D (VM5a) model (Peltier et al., 2018) is used to correct the GravIS Level-2B products for GIA. Other GIA models from, e.g., recent experiments with the VILMA code developed at GFZ (Klemann and Martinec, 2011; Bagge et al., 2021) are under consideration for upcoming GravIS versions or releases. Note that GIA model errors are not propagated into the uncertainties of the Level-2B coefficients at present.

In principle, users working with the GravIS Level-2B products could add back the applied GIA correction in the spectral domain and subtract another GIA model of interest. For the sake of completeness, it has to be noted that coefficients of degrees 0 and 1 are omitted (i.e., set to zero) when applying the GIA correction during the GravIS Level-2B processing. However, things are less straightforward in the gridded domain. This is because the generation of the Level-3 products might include processing steps that implicitly also change the applied GIA correction, which then becomes non-linear. If that is the case or not depends on the geographical domain of the Level-3 products, as well as on which particular product variable is considered. For instance, a remove-restore of the GIA correction is feasible in case of the GravIS TWS (see Sect. 2.2.2) and barystatic sea-level anomaly (see Sect. 2.2.3) grids, but is not recommended to be applied to the grids representing residual ocean circulation (see Sect. 2.2.3) or ice-mass changes (see Sect. 2.2.1).

### 2.1.6 Geophysical correction of co- and post-seismic deformations from megathrust earthquakes

While changes in the gravity field related to megathrust earthquakes are an important target signal of satellite gravity missions, such signals must be removed for oceanographic and hydrologic applications to focus on water mass signatures only. For GravIS, only tectonic signals related to megathrust earthquakes of magnitude 8.8 and larger are considered, i.e., the three seismic events of (1) Sumatra-Andaman 2004, (2) Chile 2010, and (3) Tohoku-Oki 2011. For each rupture, prior information about the position and timing of the event is obtained from an earthquake catalogue. A step function in a spherical cap with a radius of 1000 km positioned at the epicentre is then fitted to all monthly solutions within the period which was also used to calculate the long-term mean. Additionally, an exponential decay function is fitted over two years following the main event as soon as a sufficiently long time series after the event is available. Those empirical estimates are subsequently subtracted from the monthly gravity field solutions to retain only water-related signals. Note that this step is not yet included in the Level-2B products, but is performed during the Level-3 processing for TWS and ocean bottom pressure (OBP) only. Similar to the removal of the S2 tidal aliasing errors, individual monthly gravity fields are no longer statistically independent after applying this correction.

## 2.2 Spatial representation: mass anomaly grids and regional averages (Level-3 products)

Global gravity fields from GRACE/-FO contain unique information about spatially divergent mass transport processes from various components of the Earth system that are typically studied in different disciplines of physical Earth sciences. Thus, mass anomalies sensed by satellite gravimetry are attributed to individual physical processes at GravIS and grouped into three distinct regimes, namely ice-mass changes in Antarctica and Greenland (Sect. 2.2.1), TWS variations over continents excluding Antarctica and Greenland (Sect. 2.2.2), and barystatic sea-level change and bottom pressure variations in the world's oceans (Sect. 2.2.3).

### 2.2.1 Ice-mass changes in Antarctica and Greenland

GravIS provides gridded ice-mass change products for the entire Antarctic ice sheet (AIS) and the Greenland ice sheet (GrIS) (Fig. 3) as well as specifically integrated time series for major drainage basins of the AIS and the GrIS (Fig. 4). Both the gridded and the basin average products are obtained from unfiltered GravIS Level-2B coefficients (see Sect. 2.1).

**Gridded products**

Gridded ice-mass variations, developed and processed at TUD Dresden University of Technology, are provided on polar-stereographic grids with a grid spacing of 50 by 50 km. The applied algorithm was originally developed and successfully used within the ESA Climate Change Initiative (CCI) projects for both AIS and GrIS. While the formal resolution of 50 km is higher than the effective resolution of GRACE/-FO, the grid format was guided by the CCI project requirements and is a compromise to make the outline of the gridded ice sheet domains resemble the ice sheet boundaries. A more comprehensive description of the algorithm and the error assessment of the products is given by Döhne et al. (2023) and Groh and Horwath (2021). For each grid cell covering the entire AIS and GrIS, tailored sensitivity kernels, i.e., averaging kernels to be used in the regional integration approach (Swenson and Wahr, 2002), are derived. Each kernel realizes a trade-off between the following conflicting conditions, which aim to minimize spatial leakage and GRACE/-FO errors: (i) Mass changes inside the cell will be correctly recovered, (ii) mass changes outside the cell will have no impact on the grid cell, and (iii) propagated errors of the GRACE/-FO solutions have minimum influence on the mass change estimate of the cell.

To solve for the SH coefficients of each sensitivity kernel, a large number of condition equations, accounting for mass changes of the ice sheet as well as of the surrounding far-field regions, need to be established. To control the propagation of the GRACE/-FO error effects, an error variance-covariance model for the GRACE/-FO monthly solutions is required. This model is expressed as an empirical variance-covariance matrix derived from the short-term month-to-month scatter of the monthly Level-2B products. The optimal weights for the conflicting conditions are chosen from a set of plausible combinations by assessing the noise level and leakage errors in the corresponding surface mass estimates. Leakage errors are derived from a range of synthetic data sets with a priori known true mass changes, mimicking mass variations in different compartments of the Earth system.

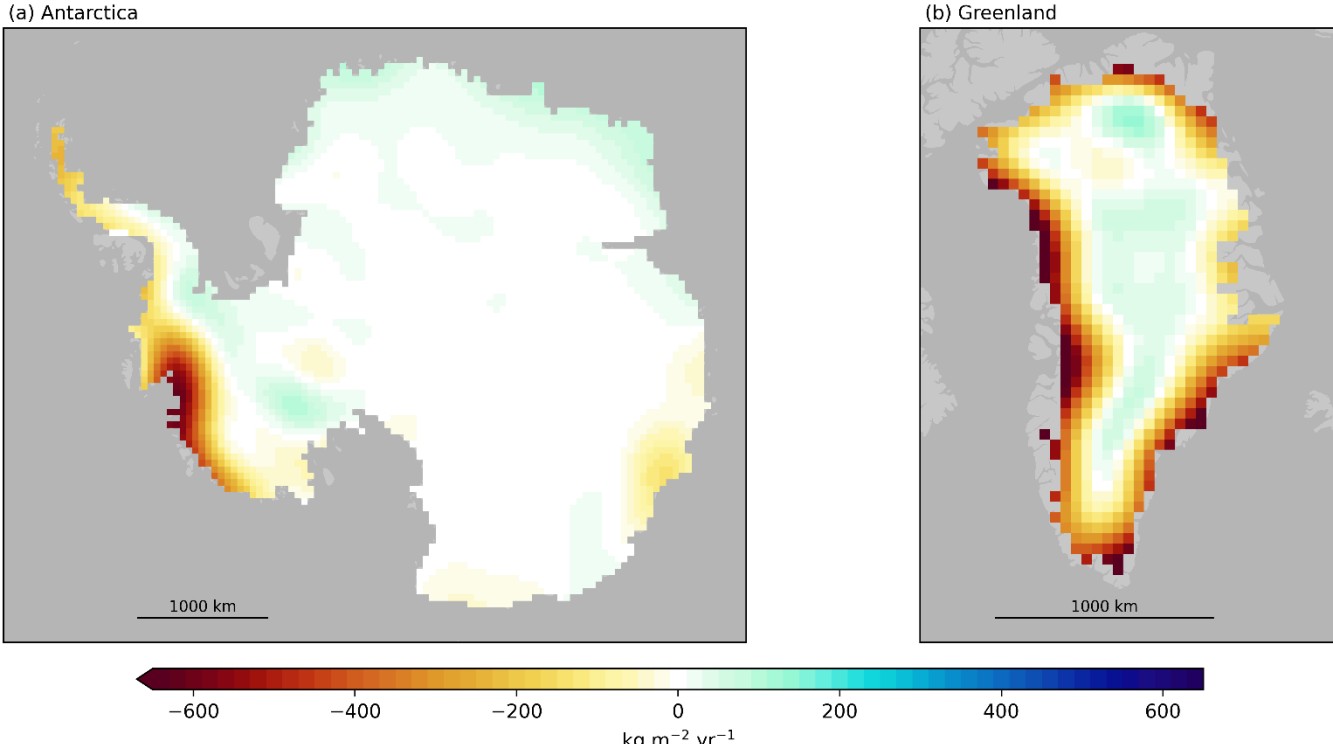

(a) Antarctica  (b) Greenland

1000 km

$\text{kg m}^{-2}\text{ yr}^{-1}$

**Figure 3: Spatial pattern of the linear mass change rate for the Antarctic (a) and Greenland (b) ice sheet from April 2002 to August 2023. The mass change rate is computed from the gridded GravIS Level-3 ice product based on COST-G.**

While error considerations guide the generation of the gridded product, definitive uncertainty measures are not part of the
gridded product but are left to the basin average products.

**Basin average products**

Basin average ice-mass variations are developed and processed at AWI. The definition of 25 major drainage basins for the AIS and seven drainage basins for the GrIS, as well as the inversion procedure based on a forward modelling approach, follows
Sasgen et al. (2013) and Sasgen et al. (2012), respectively. The inversion procedure uses predefined spatial patterns of surface-mass change of known magnitude to calculate their regional imprint in the gravity field. In a second step, the regional patterns are filtered identically to GRACE observations and least-squares adjusted (scaled) to fit the observations in the spatial domain. Using the forward model localizes the mass change more towards the coast, leading to a more realistic mass distribution with each basin compared to assuming uniform mass distribution. The inversion results are weakly dependent on the choice of the
mass distribution (less than 10 %), however, less prone to biases caused by the limited spectrum of GRACE/-FO coefficients, as both, the forward model and the GRACE/-FO data, are subject to the same post-processing procedure. For the current GravIS time series, the following processing steps are applied:

1) Spectral masking of the region of interest

2) Low-pass filtering using a Wiener optimal filter (Sasgen et al., 2006), constant over time

3) Conversion from gravity field to surface-mass changes using elastic compressible surface-load Love numbers

4) Least-squares adjustment of forward model to GRACE/-FO data

The spectral mask is set to 1 until 200 km outside the ice sheet's grounding line, following a smooth transition to 0 reached at 1000 km (AIS) or 600 km (GrIS) away from the margin. The Wiener filter is approximately equivalent to a Gaussian filter of 4° latitude spatial half-width. Along with ice-mass changes for the total ice sheet and per basin, associated calibrated 1-sigma

uncertainties are provided.

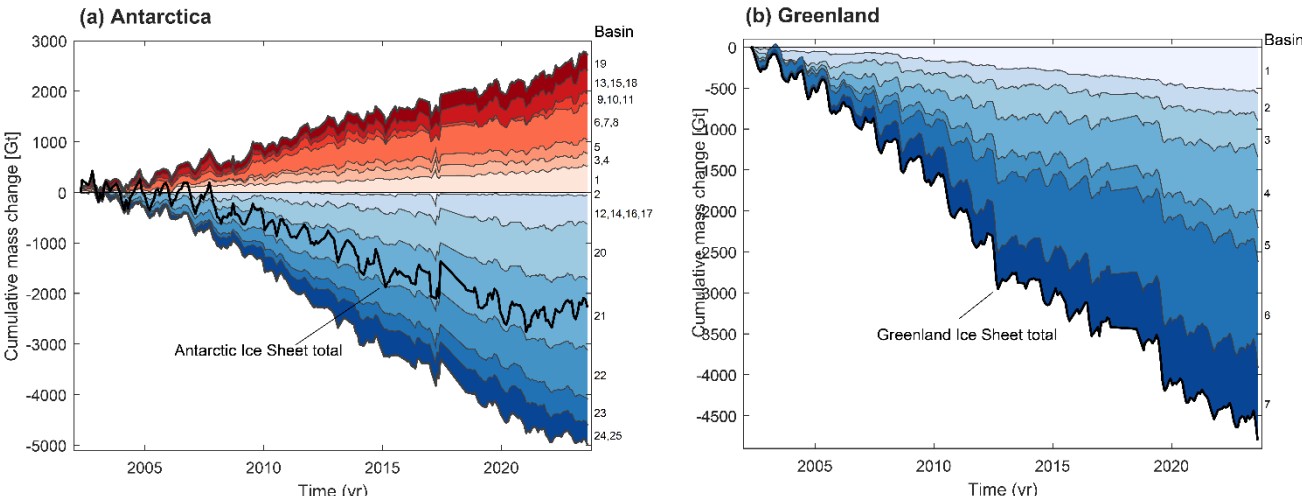

Figure 4: Time series of ice-mass change in Antarctica (a) and Greenland (b) from April 2002 to August 2023 from the basin average GravIS Level-3 ice product based on COST-G. Shown are stacked mass change time series, in blue for basins with net mass loss and

red for basins with net mass gain within the GRACE/-FO period (note that no basin in Greenland shows mass gain). The total mass change time series for each ice sheet are shown as black lines.

## 2.2.2 Terrestrial water storage variations

GRACE/-FO-based temporal changes in the Earth's gravity field over the continents are primarily interpreted as changes in

the terrestrially stored water masses. GravIS provides access to gridded as well as regionally averaged TWS products obtained from Level-2B coefficients (see Sect. 2.1).

**Gridded products**

Gridded TWS estimates, as shown in Fig. 5a, are provided on a 1° latitude-longitude grid over all continental regions except
for Antarctica and Greenland. The reference surface is the best-fitting Earth ellipsoid, as defined in the IERS Conventions (2010) (Petit and Luzum, 2010). The TWS grids contain four different variables providing monthly (1) gravity-based TWS, (2) gravity-based TWS uncertainties, (3) spatial leakage, and (4) mean atmospheric mass from a background model.

Linear trend, annual and semi-annual harmonics are estimated from time series of both VDK5-filtered and VDK3-filtered Level-2B SH coefficients. Due to the lower noise level of the trend and seasonal signals, the deterministic components from
VDK5 are subsequently combined with the residual month-to-month and inter-annual variations from VDK3. In months where the standard deviation of the residuals is 2 times larger than the mean of the monthly standard deviations, the residual variations are taken from VDK2-filtered solutions. As already mentioned in Sect. 2.1.6, the Level-2B coefficients are additionally corrected for co- and post-seismic deformations from megathrust earthquakes. Finally, mass anomalies are unambiguously inverted from the SH coefficients using the thin layer approximation (Wahr et al., 1998). In contrast to the previously published
gridded TWS data set by Zhang et al. (2016), no re-scaling coefficients from numerical models are applied.

The TWS estimates are accompanied by associated uncertainties that take into account the varying month-to-month noise level associated with (i) the amount of available sensor data from that month which might be limited due to, e.g., satellite manoeuvres, (ii) the actual ground track pattern which might be sparse during periods of occasional short repeat orbits, and (iii) certain conditions on board the satellites such as, e.g., battery conditions impacting the thermal stability and thus the noise
level of the science instruments. These uncertainties are realized as the open ocean (distance to coast larger than 1000 km) RMS of the residual signal per time step. Thus, for each month, the TWS uncertainties do not vary in the spatial domain.

The provided spatial leakage approximation is not applied to the GravIS TWS grids but is intended to enable such a correction optionally on the users' end if desired. Here, spatial leakage is based on the scaled difference of TWS fields from VDK filter pairs as proposed by Dobslaw et al. (2020b). As each TWS grid is a compound of the deterministic signals filtered with VDK5
and residual signal filtered with VDK3, the leakage correction is also a compound of the leakage correction for VDK3 and VDK5.

It has to be noted that a certain fraction of the time-variable gravity signal picked up by a satellite gravity mission is caused by atmospheric mass variability. To avoid temporal aliasing, the 3-hourly non-tidal atmosphere and ocean de-aliasing model AOD1B (Dobslaw et al., 2017) is subtracted already during the Level-2 processing of monthly GRACE/-FO gravity fields. To
provide users with the flexibility to restore atmospheric signals, the monthly mean estimate of the atmospheric background model is provided along with the GravIS TWS products.

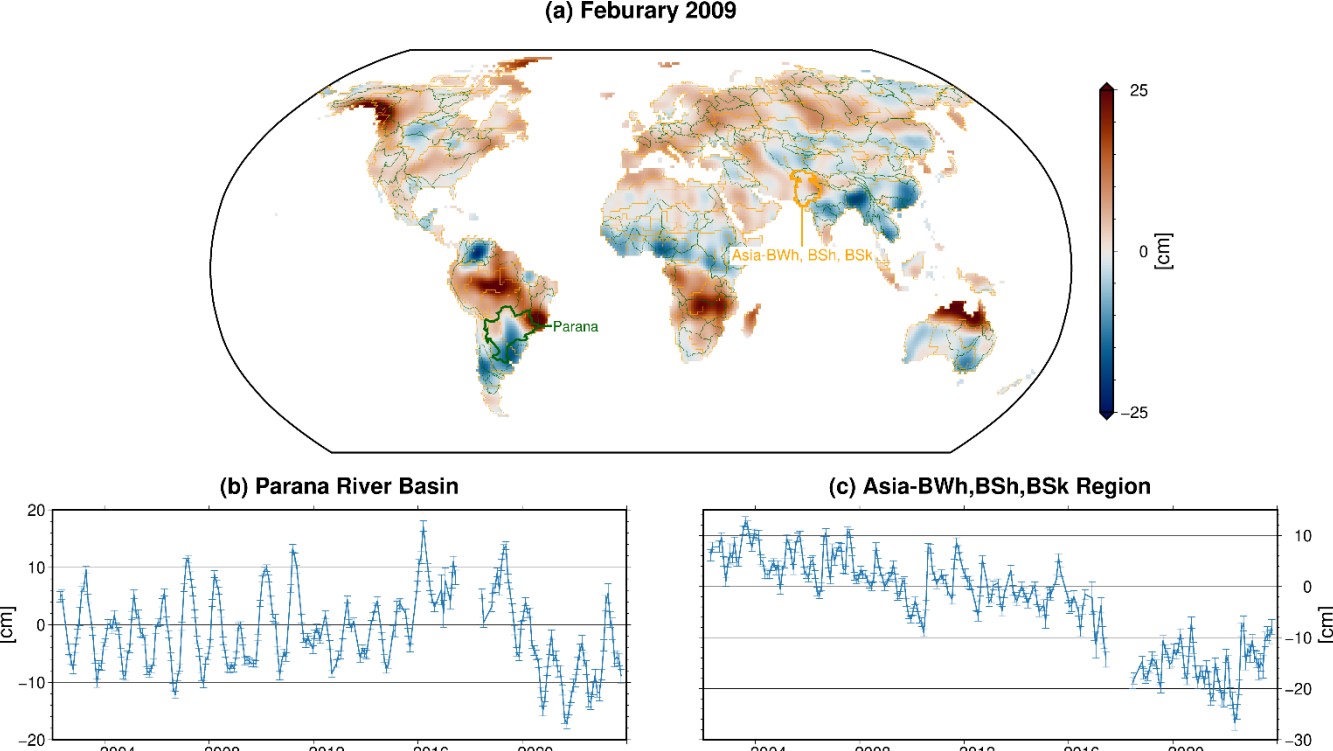

**Figure 5: (a) TWS anomalies from the gridded GravIS Level-3 TWS product based on GFZ for February 2009. (b) TWS anomaly time series from 2002 to 2023 for the Parana river basin in South America (marked with a green boundary in panel (a)). (c) TWS anomaly time series from 2002 to 2023 for a climatically similar region in northwest India and Pakistan (marked with an orange boundary in panel (a)). Visualization of all other product variables, including uncertainties and leakage, are available through the GravIS web portal (https://gravis.gfz-potsdam.de).**

**Regional average products**

In addition to the gridded products, spatially averaged TWS time series for the world's 100 largest (by area) river basins are directly available at GravIS. The total area of all these river basins covers about 56 % of the Earth's continents. To offer an alternative set of regional averages with a complete and seamless coverage of all continental regions, TWS time series for climatically similar regions as derived with a clustering algorithm (see Appendix A) are also provided. The regionally averaged TWS time series from GravIS, as shown in Figs. 5b and 5c, can be readily used for studies of terrestrial water cycle dynamics without requiring expertise in geodesy. They contain the same four variables mentioned above for the TWS grids. Yet, it is important to note that the uncertainty estimates are calculated following a modelling approach to propagate the grid uncertainties to realistic estimates for regional averages. This approach is based on a spatial covariance model which considers the non-homogeneous and anisotropic structure of spatial correlations as well as non-stationarity. Further details are described

by Boergens et al. (2020b) and Boergens et al. (2022). To allow users to compute their own regional average time series including uncertainties and covariance matrices for self-chosen regions, a Python package was published (Boergens, 2021).

### 2.2.3 Barystatic sea-level change and ocean bottom pressure variations

Satellite gravimetry, as realized with the GRACE/-FO missions, is sensitive to all mass variations in the ocean basins. Under barotropic conditions, sea-surface height changes are proportional to a change in hydrostatic pressure at the sea floor. OBP
variations are caused by the following distinctly different dynamic processes: (i) Air masses as represented by variations in atmospheric surface pressure, (ii) changes in ocean mass due to an inflow of water from the continents into the ocean basin and regional re-distribution due to attraction effects of external masses located at the continents and in the atmosphere, and (iii) the re-distribution of water within the ocean basins in response to atmospheric surface winds, atmospheric surface pressure gradients, and ocean thermohaline effects (i.e., the general ocean circulation).
GravIS provides access to both gridded and regionally averaged OBP products obtained from Level-2B coefficients (see Sect. 2.1).

**Gridded products**

Gridded OBP estimates, as shown in Fig. 6a, are provided in terms of 1° latitude-longitude grids as defined over the world's
ocean basins, again given at the Earth's reference ellipsoid as defined in the IERS Conventions (2010) (Petit and Luzum, 2010). The OBP grids contain seven different variables providing monthly (1) gravity-based barystatic sea-level pressure, (2) gravity-based barystatic sea-level pressure uncertainties, (3) gravity-based residual ocean circulation pressure, (4) gravity-based residual ocean circulation pressure uncertainties, (5) apparent gravity-based bottom pressure due to continental leakage, (6) mean ocean circulation pressure from a background model, and (7) mean atmospheric surface pressure from a background
model.

Similar to the procedure utilized for the TWS products described above, harmonics representing linear trend, annual and semi-annual signals are estimated for two variants of Level-2B SH coefficients with different VDK filters applied, here now VDK5 and VDK2. Given the less dominant annual and semi-annual signals over the ocean, the trend component from VDK5 is combined with the annual and semi-annual components and the remaining month-to-month and inter-annual variations from
VDK2. Again, similar to TWS, the Level-2B coefficients are additionally corrected for co- and post-seismic deformations from megathrust earthquakes before mass anomaly inversion (see Sect. 2.1.6). The resulting barystatic sea-level variations contain a distinct annual variation of the global mean sea-level, a pronounced positive trend, and additional strong seasonal pattern in regions characterized by monsoon circulations in the atmosphere.

The uncertainty of the barystatic sea-level pressure is provided as the temporal standard deviation at each grid point.

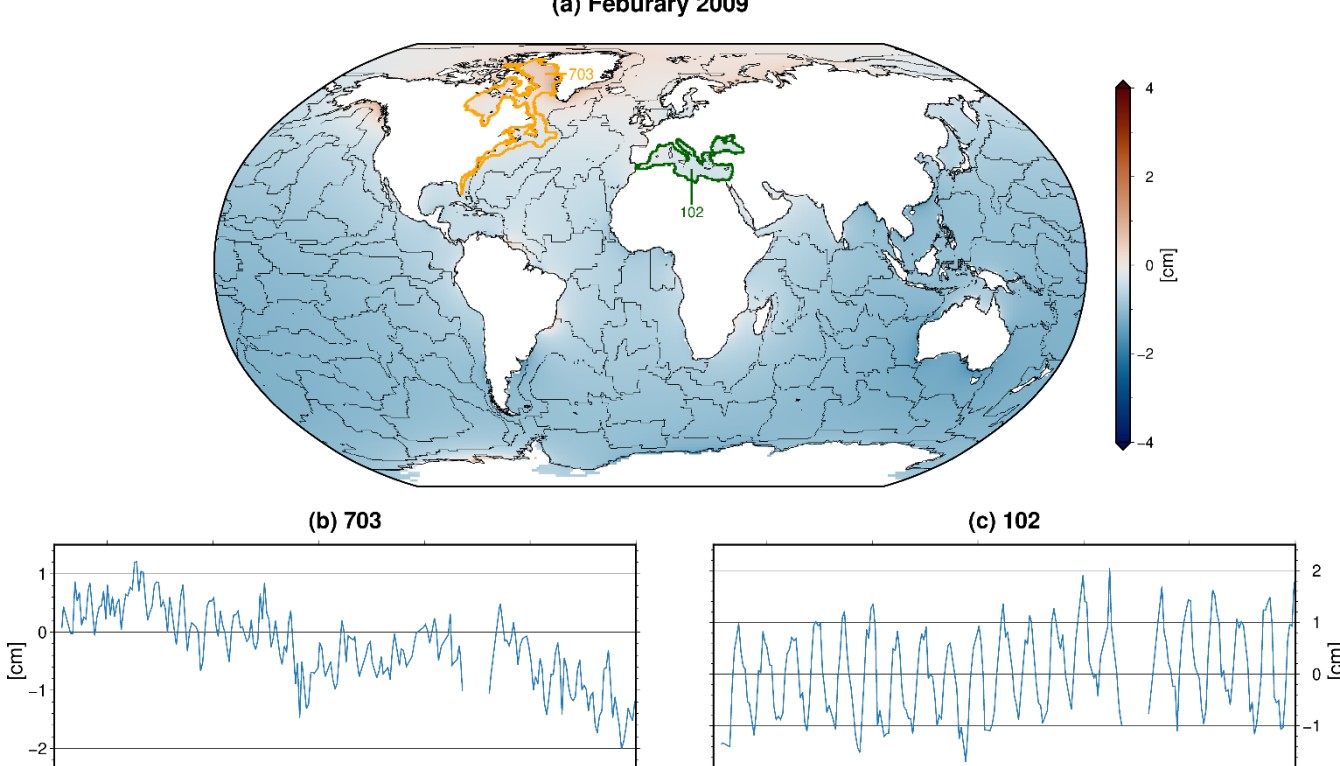

**Figure 6: (a) Barystatic sea-level anomalies from the gridded GravIS Level-3 OBP product based on GFZ for February 2009. (b) Regionally averaged barystatic sea-level anomaly time series from 2002 to 2023 for the region along the eastern coast of North America, Hudson Bay and Baffin Bay (marked with an orange boundary in panel (a)). (c) Same as (b), but for the region including the Mediterranean Sea and the Black Sea (marked with a green boundary in panel (a)). Visualization of all other product variables, including uncertainties and leakage, are available through the GravIS web portal (https://gravis.gfz-potsdam.de).**

GRACE/-FO-based TWS estimates and the associated atmospheric mass distributions from the AOD1B model are used to calculate a gravitationally consistent sea-level anomaly for each month based on the theory of Tamisiea et al. (2010). Differences between this sea-level pattern and OBP directly inferred from the Level-2B coefficients are interpreted as residual ocean circulation signals. Preliminary analysis indicates that numerous features contained in those fields are likely related to instrument noise, aliasing artefacts, or other gravity field modelling errors and thus should not be interpreted in terms of ocean dynamics. In particular, regions close to the coast are apparently affected by continental leakage.

The uncertainty of the residual ocean circulation signal is spatially constant for each time step, calculated as the standard deviation of the VDK2-filtered OBP grids reduced by the deterministic signals.

A leakage correction following Dobslaw et al. (2020b), which is a compound of OBP fields filtered with VDK2 and VDK5, is also provided. As for TWS, this correction is not yet applied but serves as an optional correction to be applied by the users, if desired.

Again, it should be noted that a certain fraction of the time-variable gravity signal picked up by satellite gravimetry is caused by atmospheric mass variability and, here in the ocean domain, also by the corresponding oceanic response to changes in, e.g., 430  surface winds. By using AOD1B, the atmospheric contribution – and to a large extent also the ocean contribution – is already subtracted during the Level-2 processing of monthly GRACE/-FO gravity fields. To provide users with some flexibility to restore those signals, the monthly mean estimates of both the atmospheric and the oceanic background models are provided along with the GravIS OBP products.

**Regional average products**

In addition to the gridded products, spatially averaged OBP time series are readily available at GravIS for climatically similar regions as derived with the same clustering algorithm used also over the continents (see Appendix A). These regionally averaged OBP time series from GravIS, as shown in Figs. 6b and 6c, contain the same seven variables as mentioned above for the TWS grids. In case of the barystatic sea-level pressure uncertainties, the values are variance-propagated from the pointwise 440  uncertainties of the gridded products, whereas in case of the residual ocean circulation pressure uncertainties, all values are the same taken from the spatially constant uncertainties of the gridded products. With the availability of such time series for individual clusters including the possibility to interactively explore them, new applications in oceanography and sea-level science might potentially be identified.

**3 The GravIS web portal**

The GravIS web portal (https://gravis.gfz-potsdam.de) was developed to fulfil two main needs in conjunction with the mass anomaly products outlined in Sect. 2: a basic description of these user-friendly products and their visualization with the possibility to interactively explore the different available products in different regions.

**3.1 Functionality and features**

Besides the portal's main page, there are dedicated subpages for the different geographical domains of the Level-3 products, 450  i.e., non-glaciated continental areas for the TWS products, oceans for the OBP products, and Antarctica and Greenland for the ice-mass change products, as well as for the corrections applied during Level-2B processing. Each Level-3 subpage provides both a zoomable map where the gridded products are displayed and a time series plot showing the regional average products. As an example, Fig. 7 shows a screenshot of the TWS subpage. If a gridded Level-3 product contains several variables, there are two ways to select a particular variable of interest: either via a drop-down menu in the map or by clicking on a variable's 455  name in the accordion menu next to the map. The accordion menu also provides a brief description of the selected variable.

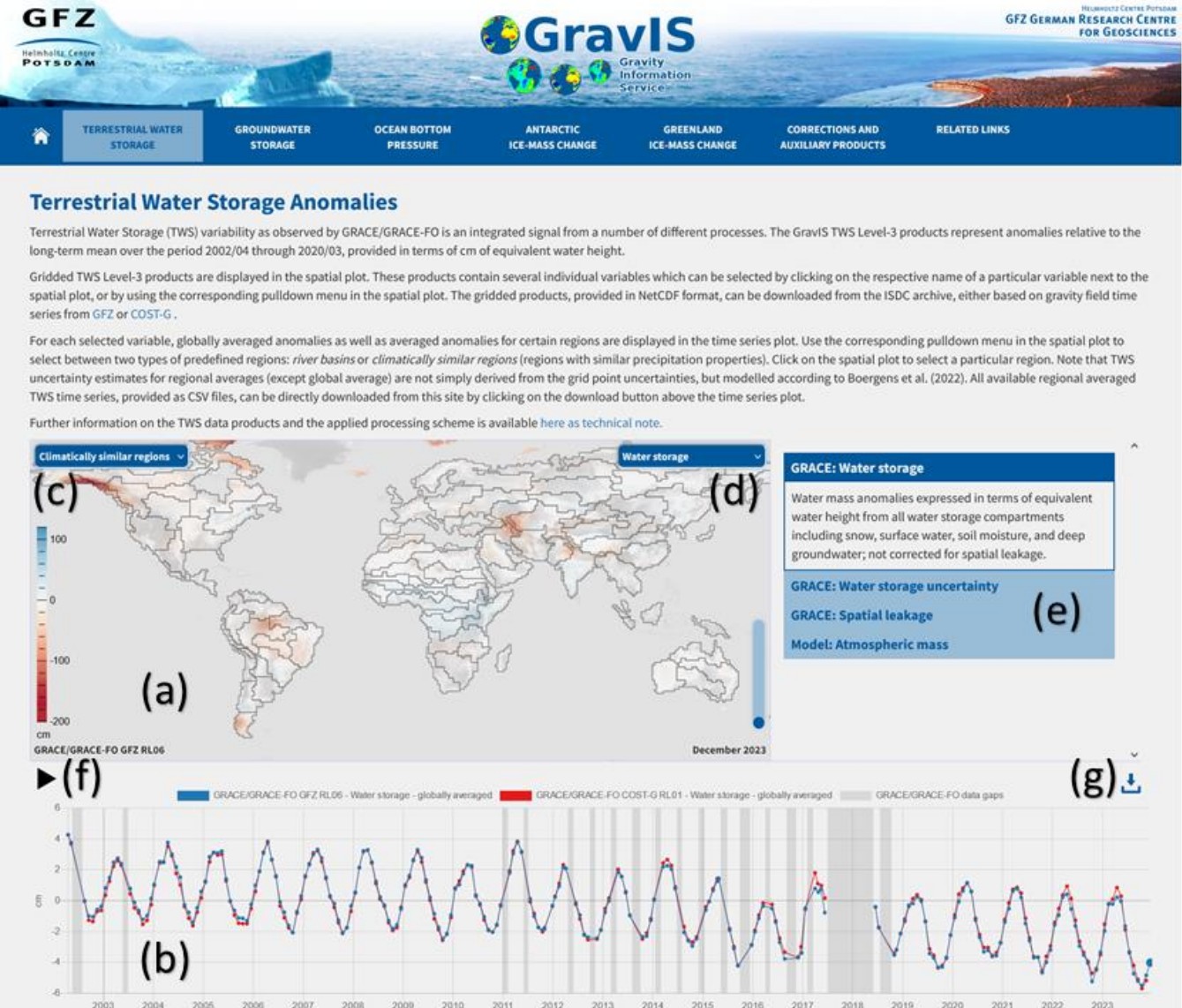

**Figure 7: Screenshot of the GravIS subpage for TWS showing the following features: (a) zoomable map, (b) time series plot, (c) drop-down menu for selection of predefined regions, (d) drop-down menu for selection of product variables, (e) accordion menu with descriptions of product variables, (f) play/stop button for time series movie, and (g) download button for regional average products.**

If more than one category of predefined averaging regions is available for a certain Level-3 product, the desired one can be selected via another drop-down menu in the map. The time series plot shows both the GFZ-based and COST-G-based products by default, but clicking on the corresponding name in the legend above the plot allows to turn a particular time series off or on again. Above the time series plot, there are two buttons: A play/stop button on the left which allows one to watch the temporal

changes of the selected variable in the spatial domain as a movie and a download button on the right to obtain the regional average Level-3 products directly from the GravIS portal. The map and the time series plot are connected with each other in several ways: (i) By clicking on a particular region in the map, the corresponding regional average time series is automatically displayed in the time series plot; (ii) by clicking on a particular data point in the time series plot, the gridded data for that exact

470 month is automatically displayed in the map; (iii) when selecting a certain product variable, it is automatically displayed in the map and as time series.

## 3.2 Complementary information at *globalwaterstorage.info*

As the focus of the GravIS web portal deliberately lies on describing and visualizing the Level-3 data sets, it is mainly addressing scientists and students who do not necessarily have to be experts in geodesy, but at least have a certain basic

knowledge about GRACE/-FO or mass change processes. Thus, it is not intended to provide more general background information about the technology and applications of GRACE/-FO on the GravIS website. On the other hand, such information is crucial for providing users with in-depth details upon request and showcasing the capabilities of GRACE/-FO to a wider audience.

To satisfy this demand, GFZ launched the information portal *globalwaterstorage.info* (https://www.globalwaterstorage.info)

which requires little to no prior knowledge, and thus nicely complements the information and data available at GravIS. This portal aims at providing easily accessible information on GRACE/-FO to the interested public, media representatives, and political decision makers. Short blog contributions cover individual aspects of the missions in easy-to-understand language. The blog covers scientific insights obtained from the mission data as well as technological aspects of the missions. It also provides regular updates on new activities related to the realization of future satellite gravimetry missions. Future blog posts

shall also focus on describing the relevance of GRACE/-FO satellite data for applications, addressing questions such as: Why is terrestrial water availability so important? Which regions of the world are particularly at risk of water loss? What developments need to be done in the future? Furthermore, the portal provides a collection of topical dossiers and fact sheets and hosts a media library collecting infographics, videos, and animations connected to GRACE/-FO.

## 3.3 Comparison with similar tools and platforms

Several other tools and websites exist, which provide, at first sight, similar functionality as GravIS by offering GRACE/-FO-based mass anomaly maps to non-expert users without requiring them to process the data themselves. In the following, a selection of a few prominent examples of such tools is briefly described and their functionality is compared against GravIS. At the *NASA GRACE Data Analysis Tool* (https://grace.jpl.nasa.gov/data-analysis-tool), users can select between the latest version of JPL and CSR mascon products, separated into land and ocean domain, and interactively analyze these Level-3 data

sets pointwise, for predefined basins, or for user-defined rectangles. Similarly, the *Mascon Visualization Tool* (https://ccar.colorado.edu/grace/) allows users to explore the mascon products from GSFC and, again, JPL. In contrast to GravIS, both tools do not offer the visualization of multiple product variables or corresponding uncertainties. Furthermore,

detailed information about the underlying gridded data sets (i.e., the mascons) and a direct download option for gridded data are not provided.

The *GRACE Plotter* (https://thegraceplotter.com/) and the *COST-G Plotter* (https://plot.cost-g.org/timeseries/) are two platforms using the same visualization tool. Various GRACE/-FO SH time series from different processing centres can be selected and compared against each other. However, limited post-processing is applied (only one type of DDK filtering and substitution of $C_{20}$ coefficients). Thus, these two platforms can be regarded as pure Level-2 visualization tools rather than a Level-3 data portal like GravIS.

With the *$G^3$-Browser* (https://icgem.gfz-potsdam.de/g3), part of the International Centre for Global Earth Models (ICGEM), there is even a further tool hosted by GFZ. Whereas the main intention of GravIS is to establish an access point for geoscientists who wish to use GRACE/-FO mass anomaly products for scientific applications, the *$G^3$-Browser* does not offer Level-3 products, but serves as a tool for educational purpose with the focus on comparing different time series for different gravity functionals. It also allows to visualize the effect of certain post-processing corrections like $C_{20}$ replacement or the subtraction

of a GIA model. In this regard, it complements GravIS rather than being a duplicate.

    In summary, there are a number of features making GravIS unique compared to other available tools: (i) It is the only portal providing regularly updated SH-based Level-3 mass anomaly products, (ii) it combines data set description, visualization, and the opportunity to download both gridded and regionally averaged products at one single place, (iii) it offers dedicated products for different applications (see Sect. 2.2), (iv) it allows to visualize different product variables including uncertainties, and (v)

it provides a proper versioning of the data sets allowing to reproduce previously published results (see Sect. 6.1).

## 4 Application of GravIS products

Satellite gravimetry has matured from a pioneering experiment to a reliable observing system for mass transport over the past two decades and is on the verge of contributing routinely to environmental monitoring services such as the U.S. Drought Monitor (Houborg et al., 2012). This is also reflected by the fact that TWS became an Essential Climate Variable (ECV) in

2020 (WMO et al., 2022). A common need for applications like this and the numerous other studies of mass change (see Sect. 1) is access to continuous time series of monthly mass anomalies derived from GRACE/-FO gravity fields. GravIS supports this need by providing routinely updated and ready-to-use mass anomaly products. Its opportunity to visually explore the data content for individual regions of interest spurs further applications and a more widespread use of satellite gravimetry data in the geoscientific community.

There are already several examples making use of Level-3 TWS products from GravIS. A prominent example is the work of Boergens et al. (2020c), who quantified the Central European Drought from space for the years 2018 and 2019. This drought continued to persist until 2023 and had severe impacts on agricultural productivity as well as water levels of major rivers in central Europe, which critically affected industrial production due to limitations in ship traffic accommodating transport of gross commodities for, e.g., the chemical industry established along the Rhine River. In another example, Liu et al. (2022)

used different GRACE/-FO-based TWS data sets, including the one from GravIS, and modelled TWS changes to quantify human-induced evapotranspiration in seven basins over the East Asian monsoon region in China. To support further hydrological applications, an observation-based prototype for a Global Gravity-based Groundwater Product (G3P; https://www.g3p.eu) was developed. G3P is based on a cross-cutting combination of TWS variations from GravIS with other individual water storage variations from existing services of Copernicus, the Earth observation (EO) component of the European Union's Space programme (https://www.copernicus.eu/en). A dedicated subpage for groundwater storage, where the latest version of the G3P data set (Güntner et al., 2024) is described and visualized, was already added to the GravIS web portal. In the near future, groundwater storage changes and TWS anomalies shall become routinely processed data sets within the Copernicus Climate Change Service (C3S). Furthermore, GravIS TWS products have been selected to contribute to the State of Global Water Resources Report of the World Meteorological Organization (WMO) since 2021 (e.g., WMO, 2023), as well as to the annual report of the *Global Water Monitor* (https://www.globalwater.online), initiating with the 2023 report (van Dijk et al., 2024).

Switching from hydrological to oceanic applications, Chen et al. (2022) compared barystatic sea level time series from GravIS Level-3 OBP products and other GRACE/-FO products including the mascon solutions from CSR (Save et al., 2016) and JPL (Watkins et al., 2015; Wiese et al., 2016). They reported a fairly good overall agreement between the different products. They concluded that slightly different trend estimates might be caused by different post-processing choices, as also shown by Dobslaw et al. (2020b).

Last but not least, also the Level-3 ice-mass change products from GravIS have been utilized. King and Christoffersen (2024) used the gridded product based on COST-G to evaluate the spatial pattern of gridded ice elevation data in Antarctica and found that both data sets are in close agreement. Graf and Pail (2023) developed a method to combine geometric and gravimetric data sets of the GrIS aiming at a higher spatial resolution of gravity-based ice-mass changes using the available GravIS Level-3 products for comparison and validation of their results.

## 5 Limitations of mass anomaly products based on satellite gravimetry

It needs to be emphasized that satellite gravimetry, as realized with GRACE/-FO, is an indirect remote sensing technique that relies on the precise monitoring of orbit perturbations of spacecraft travelling around the Earth at altitudes of about 500 km. There is no instrument pointing towards the Earth and thus, no actual sensor footprint is produced, making this mission concept distinct from other EO satellites. On the other hand, satellite gravimetry is the only remote sensing method available so far that is sensitive to mass changes. In particular, the satellite orbits are perturbed by the depth-integrated mass variations within the surface layer of the Earth, thereby making GRACE/-FO a unique tool to investigate mass changes below the surface related, e.g., to groundwater variations, deep ocean, or subglacial meltwater run-off.

However, due to this unique observing technique, it is a fundamental characteristic of GRACE/-FO that mass change estimates derived thereof are increasingly accurate when averaged over larger regions and vice versa. Averaging over larger areas both

reduces uncertainties of the observing systems and mitigates spatial leakage of continental signals, which mainly affects mass change estimates close to the coasts or along sharp horizontal contrasts in the expected mass distribution. It is thus generally advised to average GRACE/-FO results over areas of at least 100.000 km$^2$ to obtain reasonable signal-to-noise ratios. In any case, caution is advised when geophysically interpreting signals of individual 1° grid cells.

Focusing on the ocean domain, Fig. 8 shows temporal RMS values for four of the ocean products available at GravIS. It can be seen that non-tidal ocean mass variability as predicted from the unconstrained global ocean general circulation model AOD1B has signal magnitudes of several hPa (1 hPa corresponds to about 1 cm in equivalent water height) in several resonant ocean basins, particularly in the region of the Antarctic Circumpolar Current, but also in shelf regions in, e.g., Southeast Asia and the Arctic. Shelf regions are primarily affected by onshore winds leading to substantial barotropic sea-level variations with associated bottom pressure changes. Barystatic sea-level variations, on the other hand, have much smaller signal magnitudes. However, the spatial variations of mass-induced sea-level rise are much smaller, since those are only driven by attraction (and deformation) of continental mass distribution (Tamisiea et al., 2010). It also becomes visible that spatial leakage can still be significantly large in oceanic regions close to areas of substantial near-coastal ice-mass loss, as happening recently at the southern tip of Greenland and in the western part of Antarctica. Any residual signal in the oceanic mass anomalies derived from GRACE/-FO data that remains after subtraction of spatial leakage and barystatic sea-level variations is expected to reveal signatures of the general ocean circulation that are not correctly predicted by AOD1B (which is used as background model in GRACE/-FO Level-2 processing). Preliminary inspection, however, suggests that present-day signatures do not resemble bottom pressure fluctuations originating in ocean dynamics but rather processing artefacts induced by aliasing of tidal and non-tidal ocean mass variability. It is thus inevitable to continue working on improved ocean background models and refined Level-2 processing strategies to further reduce residual noise in GRACE/-FO mass estimates over the oceans.

Compared to mascon products, which are used for various studies on mass change, the approach of generating Level-3 mass anomaly products presented in this article shows some advantages and disadvantages. From a mathematical point of view, both mascons and post-processed SH coefficients are regularized solutions. Post-processing techniques, such as the VDK filter applied for the GravIS TWS and OBP products, are able to de-correlate the typical north-south oriented stripes in the GRACE/-FO Level-2 products, but also lead to signal damping and leakage, e.g., from the continents into the oceans or between neighbouring regions of interest. Typically, the applied strength of a filter is always a trade-off between sufficient reduction of noise and minimizing signal attenuation. In case of mascons, the regularization is based on certain a priori assumptions, leading to gridded mass anomaly products that look rather "clean", i.e., without exhibiting the typical stripes or signal leakage. However, due to the applied a priori assumptions, erroneous signals could be introduced or particular small-scale signals might not be correctly recovered. Moreover, mascon approaches require a better characterization of GRACE/-FO errors, which can either be derived directly from the satellite observations or from the variance-covariance information of Level-2 SH coefficients. For the combined COST-G solutions, however, the latter is currently not available. Regarding basin average ice-mass changes, mascons require to average over several grid cells, whereas the GravIS approach allows to directly analyze glaciologically defined drainage basins, accompanied by uncertainty estimates. Last but not least, the GravIS approach allows

to provide Level-2B products as an intermediate data set experienced users can work with, which is missing in case of mascon approaches.

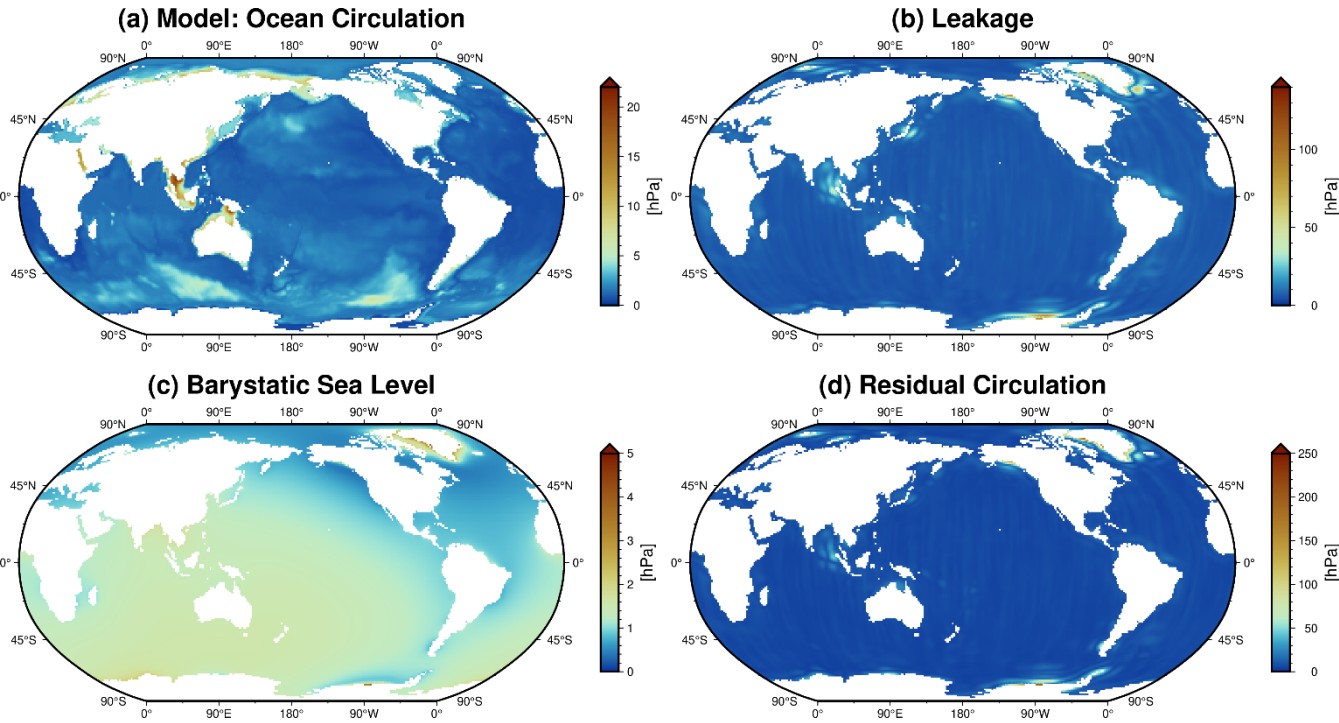

**Figure 8: Temporal RMS (2002–2021) of ocean mass anomalies for different variables of the gridded GravIS Level-3 OBP product based on COST-G: (a) mean ocean circulation pressure from a background model (AOD1B); (b) apparent bottom pressure due to continental leakage; (c) barystatic sea-level pressure; and (d) residual ocean circulation pressure.**

## 6 Data availability

The various GravIS data sets (Table 1), i.e., GRACE/-FO-based Level-2B and Level-3 products, are published via GFZ Data Services. It is important to note that all these data sets are dynamic. First, this is because GRACE-FO is still an active mission providing science data. As long as there are regular (monthly) updates of the time series of corresponding gravity field models, the time series of GravIS products will also be regularly extended. Second, quality-improved releases of GRACE/-FO Level-2 products can be expected to become available from time to time which consequently will also lead to new releases of the GravIS data sets (see Sect. 6.1 for more details regarding the update policy). Thus, the DOIs and references given in Table 1 represent the current status when this article was written and might change in the future. Note that always the most recent data sets and their citations are visualized and provided at the GravIS web portal. A comprehensive set of metadata is provided with each data set, which has been harmonized over all three data groups described in Sect. 2.2 to the extent possible.

| Data set | DOI | Reference |
|---|---|---|
| GravIS Level-2B products based on GFZ | 10.5880/GFZ.GRAVIS_06_L2B | Dahle and Murböck (2019) |
| GravIS Level-2B products based on COST-G | 10.5880/COST-G.GRAVIS_01_L2B | Dahle and Murböck (2020) |
| GravIS Level-3 Ice-Mass Change products based on GFZ | 10.5880/GFZ.GRAVIS_06_L3_ICE | Sasgen et al. (2019) |
| GravIS Level-3 Ice-Mass Change products based on COST-G | 10.5880/COST-G.GRAVIS_01_L3_ICE | Sasgen et al. (2020) |
| GravIS Level-3 TWS products based on GFZ | 10.5880/GFZ.GRAVIS_06_L3_TWS | Boergens et al. (2019) |
| GravIS Level-3 TWS products based on COST-G | 10.5880/COST-G.GRAVIS_01_L3_TWS | Boergens et al. (2020a) |
| GravIS Level-3 OBP products based on GFZ | 10.5880/GFZ.GRAVIS_06_L3_OBP | Dobslaw et al. (2019) |
| GravIS Level-3 OBP products based on COST-G | 10.5880/COST-G.GRAVIS_01_L3_OBP | Dobslaw et al. (2020a) |

**Table 1: Overview of the currently available GravIS data sets (the most recent data sets and their citations are always provided at https://gravis.gfz-potsdam.de).**

The geometries of the predefined basins and regions used for the GravIS regional average products are made available in GeoJSON format at the following links:

- Drainage basins of the AIS: https://gravis.gfz-potsdam.de/basins/AIS/antarctica.geojson
- Drainage basins of the GrIS: https://gravis.gfz-potsdam.de/basins/GIS/greenland.geojson
- River basins for TWS: https://gravis.gfz-potsdam.de/basins/rivbas/river_basins.geojson
- Climatically similar regions for TWS: https://gravis.gfz-potsdam.de/basins/clireg/land_climate_regions.geojson
- Climatically similar regions for OBP: https://gravis.gfz-potsdam.de/basins/ocean/ocean_climate_regions.geojson

**6.1 Update policy at GravIS**

As already mentioned above, all Level-2B and Level-3 data sets available at GravIS are regularly prolonged after new monthly Level-2 gravity field solutions from GFZ and COST-G, respectively, are publicly available. This extension of the GravIS time series does not imply changes of any data for previously published months; thus, no changes in the data set documentation are required. Nonetheless, there are several reasons for updating the complete time series and issuing a new version of the current
release of one or more GravIS products: (i) A new version of the underlying Level-2 products becomes available, (ii) significant methodological improvements in Level-2B or Level-3 processing have been identified, or (iii) special geophysical events requiring adapted corrections have occurred, e.g., megathrust earthquakes. In case of such a version update, the DOI of an affected GravIS data set remains unchanged, but the applied changes are documented in a changelog document. Apart from that, new releases (in contrast to new versions) of the underlying Level-2 time series will also result in a new release of the
GravIS products with different DOIs. It is worth mentioning that all previous versions and releases of the data sets with their associated documents remain accessible to ensure the reproducibility of any published results.

**7 Conclusions and outlook**

This work introduces the user-friendly GravIS mass anomaly products, including a comprehensive platform for accessing and utilizing these data sets based on GRACE/-FO satellite gravimetry data. The anomalies are attributed to different mass transport processes in the Earth system and thus are a valuable resource for related studies of mass redistribution characterized by a wide range of dynamic processes in the Earth's surface geophysical fluids and the underlying solid Earth. All mass anomaly products are routinely updated as soon as new monthly gravity fields from GRACE/-FO are available. They can be visually explored via the GravIS web portal and are provided for download, both as gridded products and regionally averaged time series, to perform subsequent scientific analysis or visualization.

Despite their limitations in spatial resolution, GRACE/-FO have been utilized with great success in numerous fields of the physical Earth sciences including hydrology, glaciology, oceanography, meteorology, geodesy, and geophysics. Providing objectively processed and user-friendly Level-3 products of mass anomalies in an interactive portal, such as GravIS, is expected to stimulate even further applications of the continuously growing satellite gravimetry time series in non-geodetic fields by lowering barriers to accessing the required data. The open data policy and the commitment to low-latency updates of the data products are designed to make GravIS a central hub for GRACE/FO-related information. In conjunction with the newly developed information portal *globalwaterstorage.info*, it will further improve the recognition of GRACE/-FO as providing an ECV in geosciences.

To further improve the information content provided with the mass anomalies from GravIS, it is intended to occasionally issue new versions or releases of the existing data products or even publish new products derived from satellite gravimetry that are particularly important for geosciences. A recent example is the G3P prototype product for groundwater storage variations based on satellite data, which was derived by subtracting estimates for the major hydrological storages based on remote sensing from GRACE/-FO-based TWS to isolate water storage changes deeper underground. Also, including near-real-time daily products based on preliminarily processed GRACE-FO data that have the potential for the rapid assessment of floods and droughts will be considered. In the future, the aim will also be to provide public and low-latency access to innovative data products from GRACE/-FO that have previously demonstrated their potential for impact in scientific communities outside the field of geodesy. Continuity in providing various mass anomaly products is ensured even beyond the mission lifetime of GRACE-FO, as GFZ and its partners have committed themselves to maintain their efforts related to GravIS in the framework of the successor mission GRACE-C (anticipated launch in late 2028) and also plan to do so in the context of ESA's NGGM/MAGIC mission launch in 2032).

**Appendix A: Climatically similar regional clusters**

The climatically similar regions mentioned in Sections 2.2.2 and 2.2.3 were identified by means of a clustering algorithm. Clustering analysis aims to group data items into subsets such that the elements within each cluster have a high degree of association among themselves and are relatively distinct from elements assigned to other clusters. As opposed to combinatorial

clustering (Hastie et al., 2009), a hierarchical approach (Ward, 1963) was applied producing a tree of clusters, where subsets at higher levels are created by merging two clusters from the next lower level.

For continental areas, monthly gridded estimates of observed precipitation as compiled by the Global Precipitation Climatology Programme (GPCC; Ziese et al., 2020) available for the period 1982 to 2019 were used. Similarity between the precipitation time series at certain grid points was measured by pairwise Euclidean distances. To arrive at clusters with rather similar area coverage, each cluster was only allowed to grow to a certain size threshold. In addition, the connectivity graph, which represents the k-nearest neighbours, was used to avoid the distribution of cluster members across multiple continents. In some rare cases, the algorithm arrived at rather stringy clusters with a low aspect ratio. To avoid such geometries, a post-processing step was introduced to merge elongated clusters with neighbouring subsets or split the clusters in half to improve their aspect ratios. Finally, a name that includes continent and climate zone, as given by the Koeppen-Geiger classification, was assigned to each cluster. In some cases, the label contains names from more than one clime zone inside the area to produce always unique names for each cluster.

The clustering algorithm invoked to derive climatically similar regions over the continents was also incorporated to disaggregate the oceans into smaller regional patches. In lieu of independently observed OBP (or the related fluxes), modelled OBP from MPIOM as simulated for AOD1B RL06 was used as the source data for the clustering. To accommodate the higher noise level over the oceans associated with the presence of residual temporal aliasing artefacts due to strong tidal and non-tidal ocean mass variability at periods shorter than 60 days that are not fully captured by the de-aliasing models, the ocean clusters are somewhat larger than those over the continents.

The geometries of the final set of clusters derived for GravIS are publicly available (see Sect. 6 for the links).

**Author contributions**

CD, HD, IS, MH, and FF contributed to the conceptualization of GravIS. SR developed and maintains the GravIS web portal. MM implemented the VDK filter method, RK developed the SLR data processing chain needed for correcting low degree SH coefficients, VK is responsible for solid Earth corrections due to GIA and megathrust earthquakes, and CD processes the Level-2B products. IS, AG, TD, and MH developed and process ice-mass change products. EB and RD are responsible for TWS data processing. HD and EB calculate OBP estimates. MS developed the clustering algorithm. US designed the information portal *globalwaterstorage.info*. CD, EB, and HD wrote the initial draft of the manuscript. All co-authors reviewed the manuscript and contributed to its revisions.

**Competing interests**

The authors declare that they have no conflict of interest.

**Acknowledgements**

EB received funding from the European Union's Horizon 2020 research and innovation programme under grant agreement No. 870353.

IS acknowledges funding from the Helmholtz Climate Initiative REKLIM (Regional Climate Change), a joint research project of the Helmholtz Association of German Research Centres (HGF).

The work of TD and AG was supported by funds from the European Space Agency through the Climate Change Initiative (CCI) projects Antarctic Ice Sheet CCI/CCI+ (contract numbers 4000112227/15/I-BB and 4000126813/19/I-NB) and Greenland Ice Sheet CCI/CCI+ (contract numbers 4000112228/15/I-NB and 4000126523/19/I-NB). The work of TD was additionally funded by the German Research Foundation (DFG) through Grant SCHE 1426/28-1.

Parts of this work were supported by funds from Deutsche Forschungsgemeinschaft (DFG, German Research Foundation) within the research unit New Refined Observations of Climate Change from Spaceborne Gravity Missions (NEROGRAV, FOR 2736).

The authors cordially thank Johann Wünsch who continuously contributes to keeping the GFZ database of GRACE/-FO-related publications up to date.

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
