# Peer review of "GravIS: mass anomaly products from satellite gravimetry"

_Earth System Science Data, 2024_

## Author Comment (AC1)

**Response to referee comments**

**Preprint essd-2024-347 by Dahle et al.**

Dear referees,

We highly appreciate the time you spent to read and review our manuscript and would like to thank you for your valuable comments that helped to improve it. Below, we have copied each comment (in italic) followed by our response to it.

Kind regards,

Christoph Dahle (on behalf of all co-authors)

**Response to referee #1:**

*The manuscript offers a thorough overview of the Gravity Information Service (GravIS), presenting mass anomaly products derived from GRACE and GRACE-FO data. The structure is well-organized, with detailed explanations of methodologies and products, and it provides substantial value by making complex satellite gravimetry data more accessible to non-experts in geodesy. This promotes broader use across various disciplines, including hydrology, oceanography, and glaciology. I would recommend publishing the manuscript in ESSD after addressing the following concerns.*

Thank you very much, we are glad to receive this positive statement.

*Major Concerns:*

*(1) The GravIS tool is described as being designed for non-expert users, offering GRACE-related products without requiring them to process the data themselves. However, there are several existing tools that offer similar functionality. The authors should provide a comprehensive list of these available platforms and compare the key features of GravIS against them. This will help clarify the unique advantages of using GravIS over other tools on the market.*

Thank you, this is a very good point. We will add a new subsection 3.3 "Comparison with similar tools and platforms" in the revised version of the manuscript. In this context, subsection 3.1 from the originally submitted manuscript will be renamed to subsection 3.2, and a large part of the original section 3 will be moved to a new subsection 3.1 "Functionality and features" without changing the content of this paragraph. The content of the new subsection 3.3 is as follows:

"Several other tools and websites exist that provide, at first sight, similar functionality as GravIS by offering GRACE/-FO-based mass anomaly maps to non-expert users without requiring them to process the data themselves. In the following, a selection of a few prominent examples for such tools is briefly described and their functionality is compared against GravIS.

At the *NASA GRACE Data Analysis Tool* (https://grace.jpl.nasa.gov/data-analysis-tool), users can select between the latest version of JPL's and CSR's mascon products, separated into land and ocean domain, and interactively analyze these Level-3 data sets pointwise, for predefined basins, or for user-defined rectangles. Similarly, the *Mascon Visualization Tool* (https://ccar.colorado.edu/grace/) allows users to explore the mascon products from GSFC and, again, JPL. In contrast to GravIS, these tools do not offer the visualization of multiple product variables or corresponding uncertainties. Furthermore, detailed information about the underlying gridded data sets (i.e., the mascons) and a download option for gridded data are not directly provided.

The *GRACE Plotter* (https://thegraceplotter.com/) and the *COST-G Plotter* (https://plot.cost-g.org/timeseries/) are two platforms using the same visualization tool. Various GRACE/-FO SH time series from different processing centres can be selected and compared against each other. However, limited post-processing is applied (only one type of DDK filtering and substitution of $C_{20}$ coefficients). Thus, these two platforms can be regarded as pure Level-2 visualization tools rather than a Level-3 data portal like GravIS.

With the *$G^3$-Browser* (https://icgem.gfz-potsdam.de/g3), part of the International Centre for Global Earth Models (ICGEM), there is even another tool hosted by GFZ. Whereas the main intention of GravIS is to establish an access point for geoscientists who wish to use GRACE/-FO mass anomaly products for scientific applications, the *$G^3$-Browser* does not offer Level-3 products, but serves as a tool for educational purposes with the focus on comparison of different time series for different gravity functionals. It also allows to visualize the effect of certain post-processing corrections like $C_{20}$ replacement or the subtraction of a GIA model. In this regard, it nicely complements GravIS rather than being a duplicate.

In summary, there are a few features making GravIS unique compared to other available tools: (i) It is the only portal providing regularly updated SH-based Level-3 mass anomaly products, (ii) it combines data set description, visualization, and the opportunity to download both gridded and regionally averaged products at one single place, (iii) it offers dedicated products for different applications (see Sect. 2.2), (iv) it allows to visualize different product variables including uncertainties, and (v) it provides a proper versioning of the data sets allowing to reproduce previously published results (see Sect. 6.1)."

*(2) To make the GravIS web portal more useful, several enhancements could be considered:*

Thank you. We appreciate the valuable and helpful suggestions on how to make the GravIS web portal more useful. However, we would like to emphasize that to our understanding, a manuscript submitted to ESSD focuses on published data sets rather than on the (further) development of possibly related web portals. Therefore, we kindly ask for your understanding that we will not consider the following suggestions within this review process. Nevertheless, we added some specific comments below to provide our point of view.

*Data Sources: GravIS currently provides mass anomaly results based on GRACE solutions from GFZ and COST-G. It would be beneficial for users to have the option to compare these results with data from other prominent centers, such as CSR and JPL. Adding a cross-comparison feature with these additional datasets would greatly enhance the utility and versatility of the service.*

As also outlined at the beginning of section 3, the idea behind the GravIS web portal is to provide a description and visualization of GFZ's user-friendly mass anomaly products. It is not intended to serve as a tool to compare different GRACE/-FO gravity field or mass anomaly products from various processing centers. Mass anomalies based on COST-G are also included at GravIS, because GFZ contributes to the COST-G service not only by providing its Level-2 gravity field time series but acts as the COST-G Level-3 Product Center (see COST-G Terms of References: https://cost-g.org/download/COST-G_Terms_of_References.pdf). For comparison purposes, GFZ hosts another tool, the $G^3$-Browser (https://icgem.gfz-potsdam.de/g3), which is part of the ICGEM service. The latter will be also mentioned in the new subsection 3.3 of the revised manuscript.

*Custom Region Selection: GravIS allows users to select predefined regions, such as river basins or climatically similar areas, for regional results. While river basin averages are commonly referenced in the literature, enabling users to define their own custom regions by drawing a polygon would add significant value. This would offer more flexibility for those focused on specific areas not covered by the existing predefined options.*

We agree. Actually, the option of customized regions is already on our list of possible medium-term enhancements to the GravIS web portal.

*Greenland and Antarctica Options: When dealing with the drainage basins in Greenland and Antarctica, only a single option is currently provided. Offering multiple options or further customization for these regions would make the tool more convenient for researchers working on these specific basins.*

Thank you for this comment. See our previous answer above, further customization of regions is considered for the future.

*Ocean Bottom Pressure: I noticed that there doesn't seem to be an option to provide results that consider self-gravitation and loading effects for ocean bottom pressure.*

GravIS provides the option to disintegrate ocean bottom pressure as sensed by GRACE into four different components. Those components include (1) atmospheric mass (as reduced with AOD1B); (2) ocean mass in dynamic response to atmospheric pressure and winds (as reduced with AOD1B); (3) barystatic sea-level variations that are gravitationally consistent with the mass distribution at the continents as observed by GRACE; and finally (4) residual ocean circulation signals observed by GRACE that are not removed with the AOD1B or contained in the barystatic sea-level variations. While AOD1B includes the feedback from self-gravitation and loading on ocean dynamics, the effect itself is not separately provided here.

*Batch Downloading: Implementing a batch download option would be extremely useful for researchers conducting global studies. The ability to download all time-series data at once would significantly streamline the process for users requiring extensive datasets for broader analyses.*

In case this idea refers to the gridded data products: For the ice-mass changes, there is already one NetCDF-file including the entire time series; for TWS and OBP, the NetCDF-files are currently divided into yearly batches, but it is already planned to also provide one file including the entire time series with the next version update.

In case it refers to the CSV-files, we note that, if no specific region is selected, these files already contain all individual regions (and all the variables). We thus think that implementing an additional batch download opportunity for a range of CSV-files does not need to be of high priority, but we will keep it in mind as an option for possible future upgrades.

*Minor concerns:*

*(1)The manuscript alternates between "GRACE/GRACE-FO" and "GRACE/-FO." It would be clearer to choose one format and consistently use it throughout the manuscript.*

In the manuscript itself, the abbreviation "GRACE/-FO" for "GRACE and GRACE-FO" is introduced in line 33 and is consistently used thereafter. "GRACE/GRACE-FO" is only used three times in the abstract. In the revised version of the manuscript, the abbreviation GRACE/-FO will be also introduced in the abstract.

*In Section 2.1.3, it appears that the authors use results based on Swenson et al. (2008), while the uncertainties are derived from Sun et al. (2016). To my knowledge, Sun et al. (2016) improved upon Swenson et al. (2008) by incorporating self-attraction and loading effects. I am curious as to why the authors chose not to use the solutions by Sun et al. (2016) directly, given that these improvements could offer more accurate results.*

Sun et al. (2016) conclude that the approach of Swenson et al. (2008) can accurately estimate geocenter variations. They do not provide geocenter solutions that could be directly used but highlight the importance of certain implementation choices for the Swenson et al. (2008) method, including incorporating self-attraction and loading effects. For the GravIS products, already existing code at GFZ is used to compute geocenter variations, which, however, lacks the capability of taking self-attraction and loading into account. This capability is planned to be added to one of the next GravIS versions. Other recommendations by Sun et al. (2016), i.e., the size of the buffer zone around the continents and the truncation degree, have already been taken into account for GravIS.

*(3)Ensure consistency of "GravIS" capitalization. It appears as "Gravis" in Line 471.*

Thanks, will be corrected.

**Response to referee #2:**

*This paper presents new GRACE/GRACE-FO mass anomaly products available on the Gravity Information Service (GravIS) portal, which includes various Level-2B and Level-3 datasets based on GFZ and COST-G. GravIS offers objectively processed and user-friendly Level-3 products (grids and regional averages) of mass anomalies for hydrology, glaciology, and oceanography applications. The authors also describe the processing steps applied to the GravIS mass anomaly products, the applications of GravIS products, and GravIS's update policies.*

*However, I have a few suggestions for improvement before the manuscript is published.*

- *Section 2.2 does not present the spatial uncertainty distribution for the Level-3 products. I believe it would be beneficial to include figures illustrating the spatial distribution of uncertainty for the three gridded products.*

In our opinion, this is a typical example showing the value and benefit of the GravIS web portal, which offers visualization in space and time for all available product variables (including uncertainties, when available). Therefore, although generally agreeing with this comment, we believe that additional figures might not be necessary in this case. Instead, we propose to add the following sentence to the caption of figure 5 and also to the caption of the new figure 6 in section 2.2.3 (see comment further below) in the revised manuscript: "Visualization of all other product variables, including uncertainties and leakage, are available through the GravIS web portal".

We would also like to state that there are no uncertainties provided with the gridded ice-mass change products. A corresponding sentence will be added at the end of section 2.2.1 "Gridded products" in the revised version of the manuscript: "While error considerations guide the generation of the gridded product, definitive uncertainty measures are not part of the gridded product but are left to the basin average products."

- *Scientists without a geodetic background can also choose mascon products provided by CSR, GSFC, or JPL. Compared to mascon data, what are the advantages and distinguishing features of GravIS products in the context of hydrology, glaciology, and oceanography applications?*

Thank you for this valuable comment. In the revised version of the manuscript, several sentences in section 1 (Introduction) will be modified or rephrased to better highlight the fact that two different approaches to derive Level-3 products are commonly used (mascon approach and Level-2 post-processing approach as used for GravIS). In addition, the following paragraph will be added at the end of section 5:

"Compared to mascon products, which are used by many scientists without a geodetic background, there are some advantages and disadvantages that are worth to be mentioned. From a mathematical point of view, both mascons and post-processed SH coefficients are regularized solutions. Post-processing techniques, such as the VDK filter applied for the GravIS TWS and OBP products, are able to de-correlate the typical north-south oriented stripes in the GRACE/-FO Level-2 products, but also lead to signal damping and leakage, e.g., from the continents into the oceans or between neighbouring regions of interest. Typically, the applied strength of a filter is always a trade-off between sufficient reduction of noise and minimizing signal attenuation. In case of mascons, the regularization is based on certain a priori assumptions, leading to gridded mass anomaly products that look rather "clean", i.e., without exhibiting the typical stripes or signal leakage. However, due to the applied a priori assumptions, erroneous signals could be introduced or particular small-scale signals might not be correctly recovered. Moreover, mascon approaches require a better characterization of GRACE/-FO errors, which can either be derived directly from the satellite observations or from the variance-covariance information of Level-2 SH coefficients. For the COST-G solutions, however, neither one or the other option would currently be given. Regarding basin average ice-mass changes, mascons would require to average over several grid cells, whereas the GravIS approach allows to directly analyze glaciologically defined drainage basins, accompanied by uncertainty estimates. Last but not least, the GravIS approach allows to provide Level-2B products as an intermediate data set experienced users can work with, which is missing in case of mascon approaches."

- *Line 126-128, the subsequent section numbers are not increasing continuously. Is there any special consideration?*

The numbered list from lines 119 to 130 (in the originally submitted manuscript) contains the different post-processing steps in exactly the order in which they are applied. In the revised manuscript, this will be better highlighted by adding "in this order" to the sentence before the list in line 118. In contrast, the following subchapters are intentionally ordered thematically (filtering followed by corrections related to low-degree harmonics followed by corrections related to geophysical signals), which causes the list's section numbers to not continuously increase.

- *In Section 2.2.3, unlike the ice-mass changes (Figs. 3 and 4) and TWS products (Fig. 5), there is no corresponding figure for the OBP product. I recommend including a diagram of the OBP gridded product along with a regional average time series.*

A corresponding figure will be added in section 2.2.3 of the revised manuscript.

- *Line 226'Currently, the ICE-6G_D (VM5a) model (Peltier et al., 2018) is used to correct the GravIS Level-2B products for GIA', and Line 228 'Note that GIA model errors are not propagated into the uncertainties of the Level-2B coefficients at present'. For users of Level-3 products, does this imply that the GIA model of gridded datasets can be adjusted simply by adding or subtracting the values from different GIA models?*

In principle, a user of our Level-2B products could add back the applied ICE-6G_D (VM5a) GIA correction in the spectral domain and subtract another GIA model of interest. However, one has to keep in mind that when subtracting ICE-6G_D (VM5a) during our Level-2B processing, coefficients of degrees 0 and 1 are omitted (i.e., set to zero).

However, in the gridded domain (Level-3 products), things are less straightforward. During the generation of the Level-3 products, some processing steps (e.g., filtering) might be included that implicitly also change the applied GIA correction, which then becomes non-linear. How much the gridded products are affected by such effects depends on the domain of the gridded products and also which particular product variables are considered.

We will think of an adequate way how this rather complex topic can be properly addressed in the revised version of the manuscript, should that be desired.

- *How is signal recovery performed during the inversion from L2 to L3, and how is signal leakage correction handled?*

Since we offer dedicated products for ice-mass changes, TWS, and OBP, signal recovery and leakage are handled differently for these different domains. We refer to the subchapters 2.2.1, 2.2.2, and 2.2.3 where, as we believe, these questions have been sufficiently answered. We also refer to chapter 5, where the topic of leakage is part of the discussion about the limitations of mass anomaly products.

- *Why is the resolution of the Gridded products set to 50 km?*

While the formal resolution of 50 km is higher than the effective resolution of GRACE and GRACE-FO, the grid format was guided by the CCI project requirements and is a compromise to make the outline of the gridded ice sheet domains resemble the ice sheet boundaries. This sentence will also be added in the revised version of the manuscript in section 2.2.1, "Gridded products".

- *Comparison: It is necessary to compare the results of spherical harmonic coefficients with Mascon products, compare with the results from different institutions, existing studies, and other measurement methods, such as satellite altimetry results.*

We agree that a comparison of the several available GRACE/-FO-based Level-3 products (mascons and post-processed Level-2 products like the GravIS products) is of general interest and relevance. On the other hand, we think that such a comparison is definitely beyond the scope of this manuscript, which is a data description paper rather than a review or research article.

- *Please provide all the codes, datasets, and results uploaded to an open-access link.*

We believe that we fulfill all requirements regarding data availability as stated in ESSD's data policy. To our understanding, authors are not obliged to provide code. If we have overseen something, we leave it up to the editor to let us know, and we will try to fulfill all required needs.